# Reemergence of Visceral Leishmaniasis in Henan Province, China

**DOI:** 10.3390/tropicalmed8060318

**Published:** 2023-06-12

**Authors:** Chengyun Yang, Suhua Li, Deling Lu, Zhiquan He, Dan Wang, Dan Qian, Ying Liu, Ruimin Zhou, Penghui Ji, Jun-Hu Chen, Hongwei Zhang

**Affiliations:** 1Department of Parasite Disease Control and Prevention, Henan Center for Disease Control and Prevention, Zhengzhou 450016, China; tochyun@163.com (C.Y.);; 2Henan Provincial Medical Key Laboratory of Parasitic Pathogen and Vector, No. 105 South Agricultural Road Zhengdong New District, Zhengzhou 450016, China; 3National Health Commission of the People’s Republic of China (NHC) Key Laboratory of Parasite and Vector Biology, National Institute of Parasitic Diseases, Chinese Center for Disease Control and Prevention (Chinese Center for Tropical Diseases Research), World Health Organization (WHO) Collaborating Center for Tropical Diseases, National Center for International Research on Tropical Diseases, Shanghai 200025, China; 4School of Global Health, Chinese Center for Tropical Diseases Research, Shanghai Jiao Tong University School of Medicine, Shanghai 200025, China

**Keywords:** visceral leishmaniasis, *Leishmania infantum*, ITS1, rK39

## Abstract

Visceral leishmaniasis (VL) was widely prevalent in Henan Province in the 1950s. Through active efforts by the government, there were no local cases reported from 1984 to 2015. In 2016, local VL cases reoccurred, and there was an increasing trend of VL cases in Henan Province. To provide a scientific control of VL, an investigation was conducted in Henan Province from 2016 to 2021. The data from VL cases were obtained from the Disease Surveillance Reporting System of the Chinese Center for Disease Control and Prevention. The rK39 immunochromatographic test (ICT) and PCR assay were performed among high-risk residents and all dogs in the patients’ village. ITS1 was amplified, sequenced, and subjected to phylogenetic analyses. A total of 47 VL cases were reported in Henan Province during 2016–2021. Of the cases, 35 were local, and they were distributed in Zhengzhou, Luoyang, and Anyang. The annual average incidence was 0.008/100,000, showing an upward trend year by year (χ^2^ = 3.987, *p* = 0.046). Their ages ranged from 7 months to 71 years, with 44.68% (21/47) in the age group of 0–3 years and 46.81% (22/47) in the age group ≥15 years. The cases occurred throughout the year. The high-risk populations were infants and young children (age ≤3), accounting for 51.06% (24/47), followed by farmers at 36.17% (17/47). The ratio of males to females was 2.13:1. The positive rates of rK39 ICT and PCR were 0.35% (4/1130) and 0.21% (1/468) in the residents. The positive rates of rK39 ICT and PCR were 18.79% (440/2342) and 14.92% (139/929) in the dogs. The ITS1 amplification products in the patients and positive dogs were sequenced. The homology between the target sequence and *Leishmania infantum* was more than 98%. The phylogenetic analysis indicated that the patients and the positive dogs were infected by the same type of *Leishmania*, which was consistent with the strains in the hilly endemic areas in China. This paper showed that patients and domestic dogs were infected by the same type of *L. infantum* and that the positive rate in dogs was relatively high in Henan Province. Because the measures of patient treatment and culling of infected dogs were not effective in reducing VL incidence in Henan Province, it is urgent to develop new approaches for the control of VL, such as wearing insecticide-impregnated collars on dogs, treating the positive dogs, spraying insecticide for sandflies control, and improving residents’ self-protection awareness to prevent the further spread of VL in Henan Province.

## 1. Introduction

Visceral leishmaniasis (VL) is an endemic infectious disease and a parasitic disease caused by *Leishmania* spp. and is transmitted by phlebotomine sandflies [1]. The main clinical symptoms are prolonged irregular fever, anemia, emaciation, pancytopenia, hepatosplenomegaly, and increased serum globulin levels [2]. Most patients die from VL-related complications within 1 to 2 years without effective treatment. As an important public health problem, VL is prevalent in 83 countries or territories across the globe [3]. Approximately 60,000-90,000 new cases of VL are reported worldwide every year, and over 90% of cases are distributed in ten countries, including Brazil, China, Ethiopia, Eritrea, India, Kenya, Somalia, South Sudan, Sudan, and Yemen [4].

In the early 1950s, VL was widely prevalent in China, mainly in 16 provinces located north of the Yangtze River [5], and it is known as one of the five most serious parasitic diseases. There were approximately 530,000 patients who were distributed throughout more than 680 counties in China in 1951, and the incidence rate ranged from 10/100,000 to 500/100,000 in different epidemic areas [6,7,8,9]. The endemic areas have been classified into three types: anthroponotic visceral leishmaniasis (AVL), mountain-type zoonotic visceral leishmaniasis (MT-ZVL), and desert-type zoonotic visceral leishmaniasis (DT-ZVL) [10,11] (Figure 1). After large-scale prevention and control, the incidence of VL declined rapidly [12]. The goal of elimination was achieved in the early 1980s [13]. Since then, only a few VLs have been prevalent in a few provinces. [14]. However, in recent years, with the development of China’s economy and the intensification of population mobility, VL has rebounded rapidly in several provinces in Central China, such as Shanxi, Shaanxi, Henan, and Hebei [15], and the epidemic scope has expanded rapidly [16].

Henan Province is located in Central and Eastern China, in the middle and lower reaches of the Yellow River, adjacent to Anhui, Shandong, Hebei, Shanxi, Shaanxi, and Hubei. The east is a plain, and the west is a mountain in Henan Province. Historically, AVL and MT-ZVL have been heavily prevalent in Henan Province. It is estimated that there were more than 40,000 VL patients in Henan Province in the 1950s [18]. Through active efforts by the government, VL was controlled, and elimination standards were basically reached in Henan Province in 1958. The last local case was reported in 1983, and there were no local cases of VL during 1984–2015 [19,20]. However, in 2016, an indigenous VL case appeared again in Linzhou City, Henan Province [21]. To date, local cases occur in Henan Province every year. In particular, the number of VL cases has increased significantly, and the epidemic area has also spread rapidly since 2020 [21,22,23]. According to our investigation of VL, it was determined that the VL cases were MT-ZVL, and the disease was mainly distributed in Northwestern Henan Province. In this study, to understand the prevalence of VL in Henan Province and to provide evidence-based data to support the adjustment of scientific strategies and measures, an investigation of humans and dogs in the villages where the patients lived was conducted in Henan Province from 2016 to 2021.

## 2. Materials and Methods

### 2.1. Data Sources and Blood Sample Collection

VL case data reported in Henan Province from 2016 to 2021 were obtained from the Disease Surveillance Reporting System of the Chinese Center for Disease Control and Prevention and investigation reports from CDCs at all levels and medical institutions. The local and imported cases were defined based on epidemiological investigations of individual cases conducted by county-level CDCs. Blood samples were collected from the patients, the people who were the patients’ family members or neighbors, and all the dogs, including domestic and stray dogs, that lived in the villages where the patients lived. Blood samples (3 mL) were collected using 5 mL EDTA blood collection tubes. The tubes were kept at 4 °C until analysis. Written informed consent was obtained from all participants or the parents of those aged less than 18 years. The study was approved by the Ethical Review Committee of Henan CDC and National Institute of Parasitic Diseases, Chinese Center for Disease Control and Prevention.

### 2.2. rK39 Strip Test

Blood serum (20 µL) was used for the rK39 ICT (Kalazar Detect^TM^ Rapid Test; In Bios International, Inc., Seattle, WA, USA) according to the manufacturer’s instructions. The results were observed in 5–10 min.

### 2.3. DNA Extraction and kDNA PCR

All patient samples were collected before treatment. Genomic DNA was extracted using QIAamp DNA blood mini kits (Qiagen, Hileden, Germany) following the manufacturer’s instructions. The obtained DNA was kept at −20 °C until further use. kDNA PCR was performed as described [24] using primers K13A (5′-GTGGGGGAGGGGCGTTCT-3′) and K13B (5′-ATTTTACACCAACCCCCAGTT-3′). The PCR system (20 μL) was as follows: 10 μL of 2 × Go Taq Green Master Mix (Promega, USA), 0.6 μL each of the primers K13A and K13B (10 μM), 2 μL of DNA template, and 6.8 μL of ddH_2_O. The conditions were as follows: 94 °C for 3 min, 94 °C for 30 s, 55 °C for 30 s, 72 °C for 1 min, 30 cycles, and 72 °C for 5 min. The expected size of the product was approximately 120 bp. Five microliters of product were detected by 2% agarose gel electrophoresis.

### 2.4. ITS1 Amplification, Sequencing and Phylogenetic Analysis

The ITS1 gene was amplified as previously described [25]. PCR was performed in a 50-μL reaction volume containing 17 μL of ddH_2_O, 1.5 μL each of the primers LITSR (5′-CTGGATCATTTTCCGATG-3′) and L5.8S (5′ -TGATACCACTTATCGCACTT-3′) (10 μM), 25 μL of 2 × Go Taq Green Master Mix, and 5.0 μL of the DNA template. The PCR conditions were 94 °C for 3 min, followed by 35 cycles at 94 °C for 30 s, 52 °C for 30 s, 72 °C for 1 min, and a final extension at 72 °C for 5 min. The expected product was approximately 320 bp. The products were sequenced by Sangon Biotech Co. Ltd. (Shanghai, China).

ChromasPro (Version 1.5) was used to splice the sequences. BLAST (National Center of Biotechnology Information, NCBI) was used to identify published sequences homologous to the target sequence and to determine *Leishmania* species. MEGA7.0 software was used to analyze the sequence alignments and construct a phylogenetic tree using the adjacency method (NJ method).

## 3. Results

### 3.1. General Status

A total of 47 *Visceral leishmaniasis* (VL) cases and 1 death were reported in Henan Province during 2016–2021. The annual average incidence was 0.008/100,000, with one death, and the fatality rate was 2.12%. The annual incidence fluctuated from 0.002/100,000 to 0.016/100,000, and the highest incidence was 0.016/100,000 in 2021, showing an upward trend year by year (χ^2^ = 3.987, *p* < 0.05). The majority of patients (87.23%, 41/47) had typical clinical symptoms, such as fever, pancytopenia, and splenomegaly. The longest time and the median time from onset to a definitive diagnosis were 546 d and 18 d, respectively.

### 3.2. Demographic, Temporal, and Geographic Distribution

The age range was 7 months to 71 years, and 44.68% (21/47) of cases were in the age group of 0–3 years, 8.51% (4/47) of cases were in the age group of 4–14 years, and 46.81% (22/47) of cases were in the age group ≥15 years (Table 1). 

Infants and young children were at a higher risk than adults, accounting for 51.06% (24/47) and 48.94% (23/47), respectively (Figure 2). The ratio of males to females was 2.13:1.

VL can occur throughout the year, but it showed two peaks in 2016–2021. The peaks were March to May and November, respectively (Figure 3).

Of the 47 cases, 35 patients (accounting for 74.47%) were indigenous cases, and the other 12 cases were imported. Thirty-five indigenous cases were distributed in Zhengzhou (twelve), Luoyang (four), and Anyang (nineteen) (Figure 4).

### 3.3. Population Infection

The blood samples of 47 patients were positive by rK39 and kDNA PCR (Figure 5). A total of 1130 human blood samples were collected in the villages where 35 local cases were located during 2016–2021. Anti-*Leishmania* antibodies were detected in all samples by rK39, four of which were positive, and the positive rate of the serum antibody was 0.35% (4/1130). The four patients had no symptoms. A total of 468 samples were detected by PCR, and the results indicated that 1 was positive. The positive rate was 0.21% (1/468).

### 3.4. Canine Infection

A total of 2342 blood samples were collected from dogs and stray dogs, 440 of which were positive for the serum antibodies. The positive rate of serum antibodies was 18.79% (440/2342). The positive rate of serum antibodies in dogs in Zhengzhou City was 30.65% (267/871); in Luoyang City, was 10.90% (57/523); and in Anyang City, was 12.24% (116/948) (Table 2). There was no significant difference in the positive rate of the serum antibodies of dogs between Luoyang and Anyang (χ^2^ = 0.4594, *p* = 0.4979), but there were significant differences between Luoyang and Zhengzhou (χ^2^ = 70.3863, *p* = 0.0000) and Anyang and Zhengzhou (χ^2^ = 91.5350, *p* = 0.0000).

DNA from a total of 929 dog blood samples was tested by PCR, and 139 dogs amplified the kinetoplast target genes of *Leishmania* spp., with a positive rate of 14.92% (139/929). One hundred and thirty dogs were positive for the antibodies and PCR simultaneously.

The information for 828 dogs was collected, such as gender, age, appearance, and symptoms. The information and the results of the serum antibodies of the dogs are shown in Table 3. There was no difference in the positive rates of gender and age, while there were significant differences in the symptoms.

### 3.5. ITS1-PCR and Sequencing Results

An approximately 320-bp fragment was amplified by PCR for all patients and 28 positive dog samples (Table 4 and Figure 5). The ITS1 amplification products were sequenced and spliced. A sequence homology analysis using MegaAllign showed that the homology between the target sequence and *Leishmania infantum* sequence in patients and positive dogs was more than 98%. Eight dogs and thirteen patient sequences were selected to construct a phylogenetic tree using *Trypanosoma* as the outgroup. The results showed that the strains of dog and patient samples were consistent with the strains in the hilly endemic areas in China and clustered into one group with *Leishmania infantum* and *Leishmania donovani* on the phylogenetic tree, indicating that the patients and the positive dogs were infected by the same type of *Leishmania*. Additionally, the sequences from the patients and dogs belonged to the same cluster, offering further evidence that the infection of the patients and dogs was related (Figure 6).

## 4. Discussion

As one of the neglected tropical diseases, VL is still a public health problem endangering global health [26]. At present, approximately 200 to 500 VL cases are reported in China every year [27], which are distributed throughout different rural areas. In recent years, the incidences of DT-ZVL and AVL have subsequently decreased, but MT-ZVL has shown an increasing trend, which is related to the resurgence of VL in some central and western provinces [28], such as Shanxi, Henan, and Hebei Provinces. There might be three factors that have caused the apparent increase in VL cases. Firstly, the terrain, landscape environment, and climate of these provinces are suitable for the breeding of sandflies. The acceleration of urbanization has also led to an increase in vacant houses and the density of sandflies. Secondly, the decline in the rural population has led to increasing dogs in rural areas for safety, and the majority of dogs are kept free, which increases the chances of dogs being infected with *Leishmania* and strengthens VL transmission. Thirdly, patient treatment and the culling of infected dogs were the main control strategy in MT-ZVL endemic regions, and these measures are not effective in reducing VL incidence. 

Henan Province was historically an epidemic area of VL. No VL cases had been reported for 33 years. In 2016, VL reappeared in Henan Province. Since then, the number of VL cases has been increasing, and the distribution has been expanding in Henan Province. The objective of this study was to provide evidence-based data and a scientific basis for the prevention and control of VL by investigating the epidemic spots of VL in Henan Province.

Herein, investigations of the epidemic spots of VL were performed in Henan Province during 2016–2021. Scientific investigation confirmed that the cases were MT-ZVL, and Henan was an epidemic area of canine-type VL. All 47 VL cases were reported in Henan Province from 2016 to 2021, including 35 local cases that were distributed throughout eight counties in Zhengzhou, Luoyang, and Anyang. The cases were scattered and had no obvious connections with each other. Of all cases, 44.68% belonged to the age group of 0–3 years, and 46.81% were in the age group ≥15, which was consistent with the investigation of VL cases in Hancheng City, Shaanxi Province reported by Han et al. [29] and the age distribution of MT-ZVL in China reported by Zhou et al. [30]. The positive rate of serum antibodies in the population at the epidemic point was 0.35%, which was far lower than the positive rate in Hancheng City [29]. Antibody-positive dogs could be detected in all epidemic spots. The average positive rate of serum antibodies in dogs was 18.79%, which was similar to the rate reported by Wang [31] in Wenxian County, Gansu Province but far lower than that of 34.3% in Hancheng City [29]. The positive rate of PCR in people and dogs was also far lower than that reported by Bai et al. [32] and Yu et al. [33] in Wenxian County, Diebu County, and Tanchang County, Gansu Province. This might be because Gansu Province and Hancheng City were epidemic areas of VL. The number of recessive infections was large, and the rate of dogs carrying leishmaniasis was high, while Henan Province had no cases for more than 30 years. Therefore, the antibody level in the population was low, and there were fewer recessive infections. 

In this study, we did not investigate the infection of Leishmania in cats, rabbits, and other animals. Foreign reports indicated that the infection of Leishmania in cats was relatively high [34,35,36,37], and cats have been shown to be important hosts for Leishmania in areas of Brazil, where the disease has been historically endemic [38]. However, there have been few reports on the infection of Leishmania in cats and rabbits in China, and most investigations have been conducted on dogs, with a few investigating domestic livestock such as cows, horses, donkeys, etc. The infection of those animals has been much lower than that of dogs [32]. We will investigate other potential Leishmania reservoirs in the future.

The genus Leishmania comprises 20 species pathogenic to humans [39], and different species and subspecies cause different types of leishmaniasis with different epidemiological characteristics. Therefore, the accurate identification of leishmaniasis is crucial to select appropriate treatment measures and formulate appropriate control strategies [40]. The internal transcribed spacer I (ITS1) sequence is located between 18S and 5.8S. The length of the nucleotide sequence is moderate, containing enough genetic information that is both conservative and sequence-specific. The degree of variation in the ITS1 sequence can reflect the genetic relationships during biological evolution and can be used for the classification and identification of Leishmania [41]. In this study, the ITS1 sequences from VL patients and positive dogs shared more than 98% homology with *L. infantum*. Therefore, it was clear that the Leishmania species causing new outbreaks of VL in Henan Province was *Leishmania infantum*. The phylogenetic analysis showed that the sequences belonged to the same cluster, indicating that the patients and domesticated dogs were both infected with the same strain of *L. infantum*. Additionally, they belonged to a cluster suggesting that the Leishmania infections in patients and domestic dogs were linked. Therefore, the patients and dogs carrying Leishmania could be potential sources of infection. According to previous reports, the density of domestic dogs is highly related to the infection rate of leishmaniasis in dogs, and the infection rate of dogs is closely related to the incidence rate of VL [42]. Reducing the breeding of dogs would reduce the incidence rate. Our investigation indicated that the density of domestic dogs was high in the villages, and the infection rate of leishmaniasis in dogs was 18.79%. In order to control the transmission, breeding fewer dogs is proposed, and the culling of infected dogs is adopted at present. However, it cannot be effective because of difficulties in implementation in Henan. 

VL is transmitted from dogs to the population by sandflies, and the sandflies that transmit VL are mainly *Phlebotomus chinensis* in Henan Province. Since the emergence of local VL cases in 2016, surveillance of sandflies has been carried out in Linzhou. With the increasing number of cases, vector surveillance has been carried out in the villages where VL cases have been reported, and several surveillance points have been set up in Henan Province. The monitoring results showed that there were mainly *Phlebotomus chinensis*, *Sergentomyia khawi*, and *Sergentomyia squamirostris* in Henan Province. *P. chinensis* was detected at all epidemic foci. At present, the cases are sporadic, and the scope is gradually expanding in Henan Province. To curb the spread of VL in Henan Province, the following measures should be taken. First, research on prevention and control strategies should be strengthened. Second, better surveillance systems are urgently needed, particularly in disease focus, for more intensive control. Third, dog management should be strengthened. It is necessary to try new prevention and control methods, such as utilizing insecticide-impregnated collars on dogs and treating sick dogs. Fourth, health publicity and education should be strengthened to improve residents’ self-protection awareness. Fifth, spraying insecticide for sandflies control should be widely carried out in areas with a high density of sandflies.

## 5. Conclusions

This paper investigated the cases and epidemic focus of VL in Henan Province. rK39 strip test results, PCR, and phylogenetic analyses indicated that the patients and domestic dogs were infected by the same type of *L. infantum* and that the positive rate in the dogs was relatively high in Henan Province. Thus, dogs and patients carrying Leishmania could be potential sources of infection. At present, the measures of patient treatment and culling of infected dogs are effective in reducing VL incidence in Henan Province. It is urgent to develop new approaches for the control of VL, such as dogs wearing insecticide-impregnated collars, treating dogs, and spraying insecticide for sandflies control. At the same time, to improve residents’ self-protection awareness is important to prevent the further spread of VL in Henan Province.

## Figures and Tables

**Figure 1 tropicalmed-08-00318-f001:**
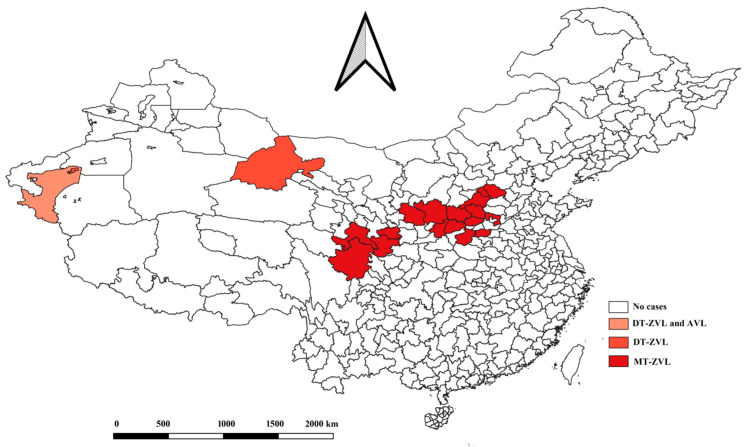
The distribution of three types of visceral leishmaniasis in China, 2021 Data cited in reference [17].

**Figure 2 tropicalmed-08-00318-f002:**
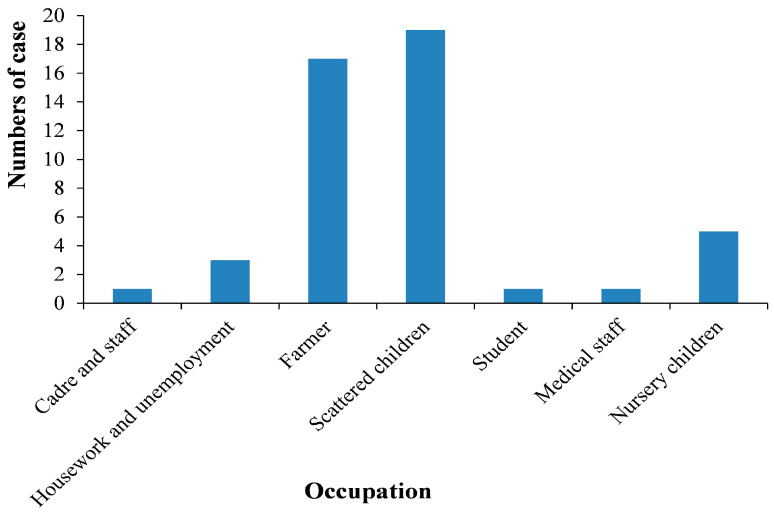
Occupation distribution of Visceral leishmaniasis in Henan Province, 2016–2021.

**Figure 3 tropicalmed-08-00318-f003:**
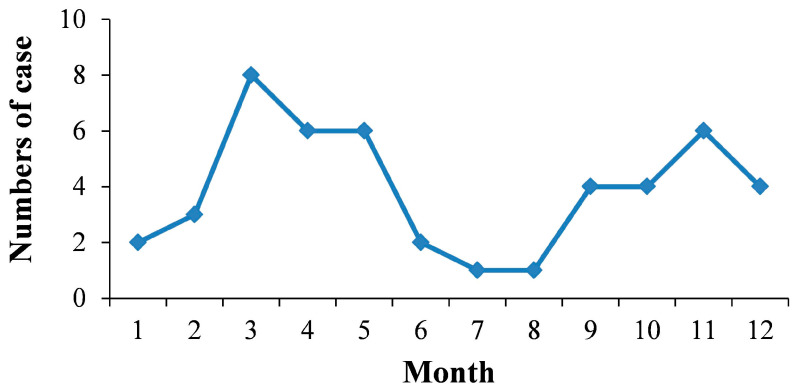
Monthly distribution of Visceral leishmaniasis in Henan Province, 2016–2021.

**Figure 4 tropicalmed-08-00318-f004:**
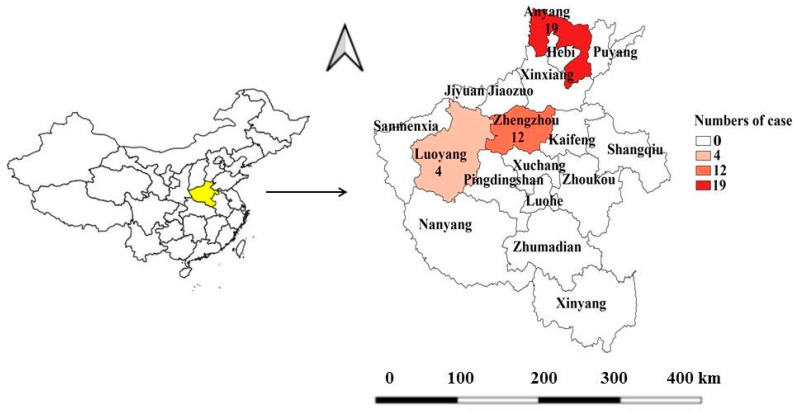
Regional distribution of Visceral leishmaniasis in Henan Province, 2016–2021.

**Figure 5 tropicalmed-08-00318-f005:**
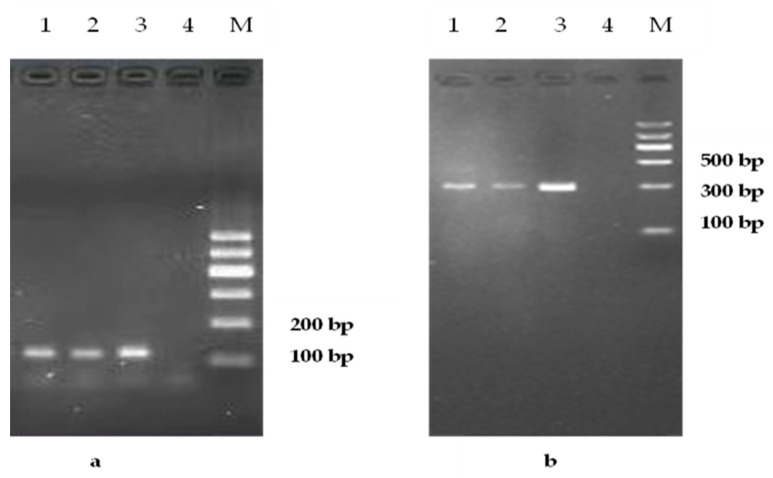
PCR products of primers K13A/K13B, 120 bp (**a**) and LITSR/5.8S, 320 bp (**b**) of *Leishmania* in the patients and positive dogs. (1) Patient blood samples of *Leishmania*, (2) positive dog blood samples of *Leishmania*, (3) positive control, and (4) negative control. M: DNA marker.

**Figure 6 tropicalmed-08-00318-f006:**
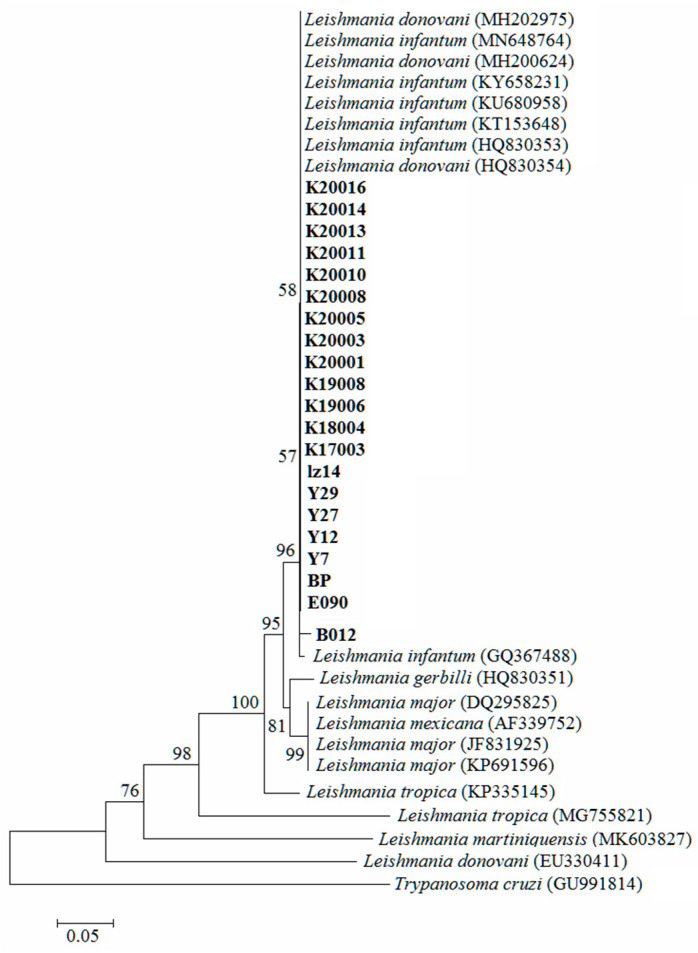
Phylogenetic analysis of ITS1 in *Leishmania* using the neighbor-joining method, with 1000 bootstrap replicates using MEGA7. K20016, K20014, K20013, K20011, K20008, K20010, K20005, K20003, K20001, K19008, K19006, K18004, and K17003, blood from patients; Lz14, Y29, Y27, Y12, Y7, BP, E090, and B012, blood from dogs.

**Table 1 tropicalmed-08-00318-t001:** Age distribution of Visceral leishmaniasis in Henan Province, 2016–2021.

Age Group (Years)	Sex		Total	Constituent (%)
	Male	Female		
0–	13	8	21	44.68
4–	2	2	4	8.51
15–	0	0	0	0.00
20–	0	0	0	0.00
30–	2	0	2	4.26
40–	6	3	9	19.15
50–	4	0	4	8.51
60–	5	2	7	14.89
Total	32	15	47	100.00

**Table 2 tropicalmed-08-00318-t002:** *Leishmania* antibody test results of reservoir host dogs in Henan Province.

City	No. Dog Investigated	No. Positive	Positive Rate/%
Zhengzhou city	871	267	30.65
Luoyang city	523	57	10.90
Anyang city	948	116	12.24
Total	2342	440	18.79

**Table 3 tropicalmed-08-00318-t003:** Factors of the dogs in Henan Province.

Variable		No. Dogs	No. Positive (Rate/%, rK39)	χ^2^	*p*
Gender					
	Male	479	98 (20.46)	0.2228	0.637
	Female	349	77 (22.06)		
Age/month					
	<24	343	75 (21.87)	0.1202	0.7289
	≥24	485	100 (20.62)		
Symptoms					
Mental state					
	Active	810	164 (20.25)	15.2745	0.0001
	Weak	18	11 (61.11)		
Appearance					
	Normal	813	167 (20.54)	7.6362	0.0057
	Thin	15	8 (53.33)		
Blephar secretions					
	Normal	824	174 (21.12)	4.0443	0.0443
	Abnormal	4	3 (75.00)		
Molt					
	No	816	168 (20.69)	7.9705	0.0048
	Yes	12	7 (58.33)		
Skin disease					
	No abnormal skin	812	163 (20.07)	25.6704	<0.001
	Yes	16	12 (75.00)		

**Table 4 tropicalmed-08-00318-t004:** The test results of reservoir host dogs in Henan Province.

Test Method	No. Samples	No. Positive	The Positive Rate (%)
rK39 test	2342	440	18.79
kDNA PCRs	929	139	14.96
ITS PCRs	42	28	66.67

## Data Availability

The datasets used during the current study are available from the corresponding author on reasonable request.

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
