# Peer review of "Reemergence of Visceral Leishmaniasis in Henan Province, China"

_tropicalmed, 2023, doi:10.3390/tropicalmed8060318_

Round 1

Reviewer 1 Report

Minor:

Line 44: avoid the term: regional

Line 45: correct the wrong word Leishimania spp. with Leishmania spp.

Line 50: correct 6,0000 with 60,000

Line 68 and following: epidemiology: a China map with the distribution of the 3 types of Leishmaniasis (AVL; MT-ZVL and DT-ZVL) could be very useful for readers that are not Chinese.

Major:

Lines 179 and following (canine infection)

It is not clear:

How many dogs resulted positive to PCR test and how many dogs were positive to antibody and PCR simultaneously;

There is no description of the dog population: age, lifestyle, gender, history. In an epidemiological study, this information is crucial. A table may be very useful

The same regards the description of dogs with and without symptoms related to canine leishmaniosis. It is necessary to have this kind of information

In case of symptomatic dogs, it is important too to have information on the way other vector borne/parasitic diseases have been ruled out

It is very important also to know how the positive dogs were managed: anti-sand flies insecticides/repellent application, drug treatment for positive/symptomatic dogs

There are no information about the presence of other potential Leishmania reservoirs like cats and rabbits, please add this information

No comments

Reviewer 2 Report

Manuscript by Yang et al. examines re-emergence of VL in Henan Province in China. The authors examined data from China CDC surveys and performed serological screening in human subjects and dogs. The results showed that the incidence of VL is on the raise, dogs are the major reservoir and the major species infecting both human and animals was L. infantum.

Overall, the manuscript is well-written and provides some data that might be of interest to local health authorities and China CDC. No explanation is provided as to the reasons for the increasing incidence of VL in China, which seemed to be effectively controlled for over 30 years. I suggest a short discussion of possible factors behind the apparent increase in VL cases.

Reducing dog breeding and overall dog management (it is not clear what the authors mean by this, presumably culling of infected dogs?) seems impractical. This has been attempted in Brazil and proved ineffective long term. Canine VL vaccination and insecticide treatment are deemed to be more effective. However, the authors do not consider the former option in their overall conclusions.

Line 53-87 this part of introduction should be shorten, it reads like a geography textbook and does not add much to the manuscript. I suggest focusing on Henan Province.

Line 50 – change 6,0000 to 60,000

Line 142 – change major to majority of

Line 150 – it is odd that infants and children are classified as occupation, this is an age group. Farmers should be classified as adults in this case.

Line 167 – this statement is confusing. Were all samples positive by rK39 test or just 4? Is it meant to say that all samples were tested? Please clarify.

Line 249 – change infantis to infantum

Reviewer 3 Report

 There are major concerns, which need to be addressed before publication.

Fig 1 was unclear; the resolution and quality of all the figures should be improved.

There is no need to mention the bands of DNA markers in the caption; write them along the band in the figures of gel agarose.

Please write the length of PCR products in the caption.

They should mention the number of positive samples from humans or dogs used in phylogenetic trees.

The bootstrap and the method used for the phylogenetic tree should be mentioned in the caption. Also, the figure is unclear; please edit it and bold the sequences from the current study.

Line 192, mention the exact number of samples.

Line195, mention the number of samples used for the phylogenetic tree.

How did they differentiate Leishmania infantum from L. donovani?

Please omit some samples/isolates in the text, you should address the number.

In the phylogenetic tree, the outgroup was missing; please correct it.

 What is the outgroup for the phylogenetic tree? Please mention it in the text.

 In a Table, please address the positive rate of ITS and kDNA PCRs and serology test.

 As the rk39 rapid test has low sensitivity and specificity in asymptomatic cases, especially in dog serum, it would be better to use DAT for seroepidemiological study. 

Round 2

Reviewer 1 Report

Thanks for accepting the proposed suggestions

Author Response

We appreciate your efforts and carefulness. Thank you very much!

Reviewer 3 Report

Figure 5b, ladder bp should be edited. 

Author Response

Dear Professor:

We have edited the ladder bp of figure 5b in our revision. Please see the revision. We appreciate your efforts and carefulness. Thank you very much!

Kind regards,

Chengyun Yang